# Daily Moderate-to-Vigorous Activity of Native Hawaiians and Pacific Islanders and Seven Asian Subgroups by Types of Activities, American Time Use Survey, 2010–2019

**DOI:** 10.3390/healthcare12020205

**Published:** 2024-01-15

**Authors:** James Davis, Deborah A. Taira, Eunjung Lim, John Chen

**Affiliations:** 1Department of Quantitative Health Sciences, John A. Burns School of Medicine, 651 Ilalo Street Honolulu, Honolulu, HI 96813, USA; lime@hawaii.edu (E.L.); jjchen@hawaii.edu (J.C.); 2Daniel K. Inouye College of Pharmacy, University of Hawai‘i at Hilo, 722 South Aohoku Place, Hilo, HI 96720, USA; dtjuarez@hawaii.edu

**Keywords:** Native Hawaiians and Pacific Islanders, Asian subgroups, moderate-to-vigorous activity, sports and recreation, household activities, American Time Use Survey

## Abstract

The study used the American Time Use Survey data from 2010 to 2019 to compare the daily moderate-to-vigorous activity of Native Hawaiians and Pacific Islanders (NHPI) and seven Asian ethnic subgroups. Adults aged 24 years and older were included. The study analyzed activities from sports and recreation, household activities, and all activities carried out during the day. Outcomes were determined by the completion of 30 min or more of moderate-to-vigorous activity and the type of activity carried out in the day. Significant ethnic differences were observed for sports and recreation but not for household activities and not for all activities carried out during the day. Of the ethnic populations, NHPI were the least active, and Asian Indians and Chinese were the most active. A majority achieved 30 min or more of moderate-to-vigorous activity during the day from all their activities. Physical activity from household activities exceeded physical activity from sports and recreation. The most physically active group was adults over the age of 65 years, perhaps reflecting more time to exercise or greater concerns about their health. For sports and recreation, exercising with someone doubled the minutes of moderate-to-vigorous activity. The results emphasize the importance of activities performed around the household in addition to sports and recreation and the benefit of exercising with someone. Ethnic populations may be receptive to interventions that emphasize activities they are performing in their daily lives.

## 1. Introduction

Native Hawaiians and Pacific Islanders (NHPI) and Asians are understudied minorities [1,2,3]. NHPI suffer heightened risks of hypertension, diabetes, and stroke (and experience strokes at younger ages), have shorter average lifespans, and live in poorer socioeconomic conditions [4,5,6]. The historic overthrow of the Hawaiian monarchy in 1893 forced relocation, a loss of culture, and historical trauma, leading to enduring health disparities [7]. Today’s NHPI youth still suffer from historical trauma [8]. Asians are frequently seen through the lens of the model minority myth, which holds that they are one homogeneous group that is highly successful, educated, and diligent [9]. In reality, the social conditions, histories, and cultures of Asians are extremely diverse. Asian ethnic groups range from the top to the bottom 10% of income nationally, and the aggregate average hides inequalities [10]. They face high risks of cardiometabolic abnormalities based on their obesity and fat distribution [11,12,13].

Historically, NHPIs and Asians were combined into a single Asian and Pacific Islander group, precluding the understanding of single ethnicities. In 1997, the Office of Budget and Management recommended disaggregating the Asian and Pacific Islander categories into Asians and NHPI categories (OMB 1997; Hixon et al., 2012) [14,15]. Three years later, the Department of Health and Human Services (HHS) disaggregated the Asian race category into six Asian subcategories that make up the majority of Asian Americans who identify with one Asian subcategory [16,17,18]. Disaggregation of national databases has improved in recent years although many do not meet the HHS standard of disaggregation. Analysis of the disaggregated data can inform culturally tailored interventions and health policies to improve minority health. Tailored interventions have proven effective across culturally diverse populations [19,20,21]. 

This study uses the American Time Use Survey (ATUS) to examine the amount of time NHPIs and the seven major Asian ethnic groups spend performing moderate-to-vigorous activities in sports and recreation, household activities, and all daily activities. According to the position paper of the American College of Sports Medicine, irrefutable evidence documents that even moderate amounts of physical activity are associated with a decreased risk of cardiovascular disease and early death [22]. The paper recommends a minimum of 150 min (or two and a half hours) of moderate-to-strenuous activity along with a daily goal of 30 min or more. 

The ATUS is a large national database sponsored by the Bureau of Labor that monitors the activities participants perform across a 24 h day (1440 min of activity). Activities are assigned metabolic equivalents (METs) based on published data [23]. Recent research utilizing the ATUS investigated for the national population the extent to which different sorts of activities, including leisure activities and daily tasks like housework, contribute to moderate-to-vigorous activity [24,25]. We extend these studies by examining moderate-to-vigorous activity among NHPI and seven Asian populations. 

## 2. Methods

### 2.1. Study Design

The study was designed to understand moderate-to-vigorous activities performed during the day by Native Hawaiians and seven Asian ethnicities. The number of respondents from other Pacific Islander subgroups was too small for their data to be analyzed. The ATUS recruits a random subset of households completing their eighth month of interviews for the Current Population Survey, the nation’s monthly labor force survey. The sample is nationally representative of the US population. One individual aged 15 years or older per household is randomly selected to take part in the ATUS; the person is asked how they spent their previous day, where they were, who they were with, and what tasks they performed throughout the day. The time reference goes from 4 am on the previous day up until 4 am on the interview day. The interview lasts about 20 min. By design, half of the participants report on a weekday and half on a weekend day. The ATUS is conducted using computer-assisted telephone interviewing. The interviewer uses scripted open-ended questions along with conversational interviewing, a flexible interviewing technique. Common activities such as sleeping and eating are directly coded in the computer; other activities are recorded verbatim so that experienced interviewers can later assign codes to the reported activities. The minutes taken to perform activities are recorded in chronological order across the 24 h of the previous day. 

Interviewers are trained in both interviewing and coding. They learn how to code activities reliably and to probe for detail if the respondent gives a vague response. They are trained on how to probe in a consistent manner. They are not only trained to code their own interviews but also to code other interviewers’ dairies. 

### 2.2. Study Population 

The study included all available Native Hawaiian and Asian participants in the ATUS from 2010 to 2019, aged 24 years and older. Age 24 was chosen so that participants would have time to complete their education. Native Hawaiian participants were included from 2010 and Asian participants were included from 2013. The earlier years for Native Hawaiians were included to augment their small numbers. The ATUS included 73,626 participants from 2013 to 2019. The overall percentage of NHPI was 4.4%. Individuals were excluded if the interviewer did not think the interview should be used, as indicated by a variable in the ATUS data. Interviewers noted interviews that could be excluded because respondents incorrectly remembered activities, intentionally provided wrong answers, or deliberately reported very long durations, among other reasons. A subset of 2865 participants who identified themselves as Native Hawaiians or Asians was included in the analyses. The total N for the study years averaged about 10,000 participants per year.

The ATUS data are publicly available through the Integrated Public Use Microdata Series [26]. 

### 2.3. Study Variables 

Age was categorized into five groups (24–35, 36–45, 46–55, 56–65, and over age 65). Sex was grouped as male and female. Health status was in two categories: good, fair, or poor and very good or excellent. Education was grouped as less than high school, high school, or college. Ethnicity was Native Hawaiian, Chinese, Asian Indian, Filipino, Japanese, Korean, Vietnamese, and Other Asian. The day of the week was weekday or weekend. 

The study’s outcomes are based on moderate-to-vigorous activities with three metabolic equivalents (METs) or higher. Tudor-Locke and colleagues assigned METS to 438 activities in the ATUS using the Compendium of Physical Activity [23]. The primary outcomes are having 30 min or more of moderate-to-vigorous activity from sports and recreation and household activities, as well as having 30 min or more of moderate-to-vigorous activity during the previous day based on all activities. A secondary outcome was the total minutes of moderate-to-vigorous activity during the previous day.

### 2.4. Statistical Analysis 

Initial analyses summarized the number and percentage of people by characteristics of the study population. As a second step, binary outcomes were cross-classified by participant characteristics. Subsequent analyses employed multiple regression models. Analyses with 30 min or more of moderate-to-vigorous activity as the outcome were analyzed using quasi-binomial regression. The results of the quasi-binomial models were summarized as odds ratios with 95% confidence intervals (CIs). Analyses with the total minutes of moderate-to-vigorous activity as the outcome were analyzed using quasi-Poisson regression. The results are provided as relative minutes of activity with 95% CIs. All analyses used the complex survey weights provided by the ATUS to align with the design of the ATUS. Statistical analyses were conducted using R version 4.3.1 and employed the survey package designed by Lumley for analyses of complex survey data [27,28]. 

## 3. Results

The frequencies of the study participants are given in Table 1. Native Hawaiians are listed first, followed by the other Asian ethnicities from most to least common. Chinese and Asian Indians were the most common ethnicities (25.5% and 20.3%, respectively), with Japanese and Koreans the least frequent (both 5.8% of the population). The proportions of participants by age group decrease monotonically from 34.3% for ages 24–35 to 9.4% for participants over 65. There were about 9% more females than males, and 9% more participants reported very good or excellent health than reported good, fair, or poor health. A majority reported a college education (66.4%), and weekdays exceeded weekend days.

Table 1 also provides bivariate associations with moderate-to-vigorous sports and recreation. Ethnicity, age group, sex, health status, and education were significantly associated. Of the ethnicities, Native Hawaiians had the lowest percentage undertaking sports and recreation (12.2%), and Chinese and Japanese had the highest percentages (24.0% and 21.5%, respectively). Age group, sex, education, and weekdays were significantly associated with moderate-to-vigorous household activities, which were more often performed on weekends than on weekdays (34.3% vs. 28.3%). The percentages of participants carrying out moderate-to-vigorous household activities exceeded the percentages undertaking moderate-to-vigorous sports and recreation.

Figure 1 compares 30 min or more of moderate-to-vigorous activity (1) based on all activities reported, (2) based on sports and recreation, and (3) based on household activities. The results are presented as a step curve to illustrate the changing proportions of participants. The upper steps show that more than 50% of participants had more than 30 min of moderate-to-vigorous activity from all the activities they performed during the day. The middle line shows increasing proportions with older age groups from household activities. By comparison, the lower line shows that lower proportions of participants achieved 30 min or more of moderate-to-vigorous activity in sports and recreation. For all three outcomes, people over the age of 65 had the highest activity levels. 

Figure 2 offers a more detailed look at the primary activities that participants were performing. Housework was the most common for all age groups. Lawn and garden activities were common as well, with increasing frequency at older ages. Taking care of animals was the third most popular. Walking was the predominant sport and recreation activity. Walking doubled in frequency from the youngest to the oldest age groups. Running and weightlifting had a subset of adherents, mostly in the youngest age groups.

To gain a more detailed understanding, we fit two multivariable regression models with 30 min of sports and recreation as the outcome for the first model and 30 min or more of household activities as the outcome for the second model. Table 2 compares the models. The only statistically significant associations with ethnicity occurred with sports and recreation; compared to the NHPI, Chinese and Asian Indians were more than twice as likely to partake in moderate-to-vigorous sports and recreation. Participants over age 65 were three times as likely to participate in sports and recreation as those aged 24–35 (OR = 3.06, 95% CI = 1.71, 5.49) and twice as likely to carry out household activities (OR = 1.96, 95% CI = 1.38, 2.77). People from 36 to 55 years of age were significantly more likely to engage in moderate-to-vigorous household activity than those ages 24–35. Odds ratios ranged from 1.42 to 1.78. Moderate-to-vigorous household activities increased with increasing age groups and were more frequent on the weekends than on the weekdays (OR = 1.28, 95% CI = 1.05, 1.56).

Table 3 presents the results for completing 30 min of moderate or vigorous activity from all activities combined. In cross-classified analyses, Chinese and Asian Indians had the highest proportions (58.4% and 57.7%, respectively); only Koreans with 45.1% were lower than 50%. Ethnicity, age groups, and day of the week were statistically significant. In adjusted regression models, ethnic differences became non-significant, suggesting other participant characteristics explained the ethnic differences. Only being over age 65 relative to ages 24–35 years was statistically significant in the multivariable regression model.

Table 4 presents a different perspective: analyses are limited to participants who participated in sports and recreation, using the total minutes of moderate-to-vigorous activity as the outcome. The analysis asks if exercising with someone increases the number of minutes spent performing activities. On average, exercising with someone increased the number of minutes of sports and recreation by 93% (relative ratio = 1.93, CI = 1.63, 2.29). Restricting analyses to participants participating in sports and recreation uncovered ethnic differences. Compared to the NHPI, Chinese, Asian Indians, Filipinos, and Koreans all had fewer relative minutes of activity when controlling for other participant characteristics.

## 4. Discussion 

Across the age range, a majority of the participants engaged in at least 30 min of moderate-to-vigorous activity during a given day. This finding was true for NHPI and all seven Asian subgroups. The activities came from sports and recreation and household activities, but more moderate-to-vigorous activity came from household activities. The duration of exercise increased when participants exercised with someone else rather than alone. 

Our observations are consistent with previously published studies. According to a British study of women between the ages of 60 and 79, more than two-thirds of them engaged in the recommended amount of activity [29]. If excluded, only 21% of the women achieved the recommended activity levels. This study did not find that socioeconomic status was associated with domestic activities. A second British study reported that 42.7% met recommended physical guidelines, with 35.6% reporting moderate-to-vigorous activity from domestic physical activity [30]. If domestic activity was excluded, only 20.4% of participants reached the current recommendations for activity. One study using the ATUS as we did was interested in understanding opportunities for groups with lower socioeconomic status (SES) to achieve adequate exercise [24]. The investigators found that participants with lower SES received more physical activity from housework and dependent care than their more educated counterparts. The investigators concluded that activity domains beyond leisure time physical activity should be promoted to encourage physical activity. A second study using the ATUS grouped participants’ non-labor activities (i.e., activities outside of work) into patterns and saw that housework and caregiving patterns were comparable to the exercise pattern in terms of overall energy expenditure [25]. A study of retired couples observed that people continued to perform housework except when in poor health, and if one partner was in poor health, the healthier member increased their activity to compensate [31]. Even with bad health, housework only declined by about a sixth. A study of adolescents randomized the adolescents between conventional exercise and housework-based exercise. Housework was concluded to be just as good for keeping or improving adolescent learners’ fitness [32]. Hence, our findings from NHPI and Asian survey respondents are in line with other studies that have found the important contribution of household activities to moderate-to-vigorous activity.

Participant characteristics associated with carrying out the various moderate-to-vigorous activities differed by the study outcome. Based on multiple adjusted models, the oldest population was the most physically active: being 65 years of age or older was statistically significant for sports and recreation, household activities, and overall daily activities. Older adults may have more free time or value the health benefits more. Our findings are consistent with several previous studies. Koeneman et al., 2012 reported that retired participants spent more time in moderate-to-vigorous activities than participants who were still employed [33]. Van Dyck et al., 2016 reported the level of physical activity increased around retirement age but had complex interactions with age, gender, and education [34]. Liu et al., 2017 describe, in a systematic review, the associations between physical activity and improved hypertension [35]. Foong et al., 2014 found that as age increased, the effects of light and moderate activity on adiposity decreased [36]. 

Two prior research studies used the 2014 Native Hawaiian and Pacific Islander National Health Interview Survey (NHIS) to examine physical activity and functional limitations among NHPI. According to the first study, functional restrictions surge in NHPIs in their mid-60s, whereas white people did not experience a similar surge until their mid-70s [37]. In the multivariable-adjusted regression models, NHPI participants who met recommendations for aerobic and strengthening exercises had better function using medical equipment and less difficulty walking [38]. People who met the aerobic requirements had less trouble taking care of themselves. The second study looked at achieving 150 min or more of physical exercise per week [39]. Only half of the study participants met this level of physical activity. Participants who did were less likely to experience memory issues. Our study adds to these prior studies by examining time spent in household activities for NHPI in addition to exercise and by comparing NHPI to Asian subgroups. 

In our study, ethnic differences were only apparent for Chinese and Asian Indians; they had more than twice the odds of Native Hawaiians for moderate-intensity sports and recreation, although activity from household activities did not differ significantly. Several other study variables had associations with recommended daily activity. Compared to those in good, fair, or poor health, those in good to exceptional health had 60% higher odds of undertaking moderate-to-vigorous exercise. Females performed household activities more often than men. Participants performed household activities more often on weekends than on weekdays. When analyses were restricted to sports and recreation, people who exercised with a partner increased the amount of time spent engaging in moderate-to-vigorous activity by 93%. Native Hawaiians tended to exercise longer than individuals of Asian ethnicities, and weekend exercise was generally longer than weekday exercise. Further research is needed to understand the mechanisms explaining these differences. For instance, these differences may arise from the type of exercise. 

Culturally tailored interventions have shown success in the NHPI community. A study in Southern California Pacific Islander communities showed an increase from 2.09 to 2.80 days per week of medium-intensity physical activity [21]. Fifty-two percent of participants were not physically active at baseline. The Kā-HOLO Project in Hawaii was a randomized controlled trial of a hula dance program for Native Hawaiians in Hawaii. Hula can be performed with varying intensities, but even low-intensity hula meets the recommended guideline for moderate-intensity exercise and can be prescribed for physical activity [22]. Participants in the Kā-HOLO Project were Native Hawaiians with uncontrolled hypertension. At six months, the intervention group had significant improvements in systolic (−15.3 mm HD) and diastolic (−6.4 mm Hg) blood pressure compared to controls who participated in an education program (−11.8 mm Hg and −2.6 mm HG for systolic and diastolic blood pressure). The improvements were retained at 12 months. A second Hawaii study translated the Diabetes Prevention Program Lifestyle Intervention into a 3-month community intervention of NHPI [22]. Improvement occurred for changes from pre- to post-intervention in weight, blood pressure, physical function, exercise frequency, and fat in the diet.

Our findings that the population health of Asians varied across ethnic subgroups and by years living in the US is consistent with prior studies. Compared to non-Hispanic whites, the Filipino population in the California Health Interview Survey reported a high prevalence of hypertension, obesity, and diabetes. Japanese and Chinese people in the study were more likely to be obese. Japanese people were more likely to have diabetes, Koreans were less likely to have a regular source of care, and Vietnamese people were more likely to have fair or poor health. According to a second study that used the California Interview Survey, the biggest risk factor for heart disease among Filipinos is hypertension [40]. Low levels of physical activity were reported by Asian Indians in the National Health Interview Survey (NHIS) [41]. Being born in the US or staying in the US for 10 or more years in the NHIS among Asian American adults in the United States was positively associated with meeting recommended physical activity levels [41]. The NHIS showed an increase in the popularity of Asian meditation-based mind–body activities like Tai Chi and Qi Gong, which can offer culturally appropriate, economically advantageous mind–body exercise to reduce health inequities [42].

### 4.1. Limitations

Our study has several limitations to consider in interpreting the results. One limitation of the study is that activity in the ATUS is recorded for a single day. We could study a daily exercise goal of 30 min or more of moderate-to-vigorous activity but not the weekly goal of 150 min or more based on national guidelines. Because participants are asked about the previous day rather than the current day, the information provided may have recall bias. Another limitation is that the analyses did not control for the number of hours a person spent sleeping per day. Someone who sleeps for 7 h a day would have more time to exercise than someone who sleeps 10 h a day. Remembering activities from the previous day in detail may challenge some participants, and the estimated times they were performing activities could have errors. Our reference population is Native Hawaiians. A more general population also would be of interest. As a further concern, the participants are fairly educated, suggesting a possible selection bias among ethnic minorities in the ATUS. Minorities are reluctant to join studies and those that do may not be fully representative. Although the ATUS assigns METS to all activities in a day, activities are based on self-report and could have errors. The gold standard for measuring activity is accelerometer techniques [43].

### 4.2. Strengths

The nationally representative sample of Native Hawaiians and Asian subgroups is a strength of our study. The results contribute to the still limited information on these study populations. Other strengths include that the data on METS come from a published methodology, activity is summed from the many activities, sports and recreation and household activities can be examined separately, and the types of activities are based on a published methodology. 

### 4.3. Implications 

The main factor that led to moderate-to-vigorous activity every day in our study, as well as in other populations, was completing tasks around the house. This suggests that Native Hawaiian and Asian subgroups could be more active by focusing on the things they carry out every day. Exercise programs might be individually customized to fit with a person’s daily activities. Household activities, including gardening, yard work, home repairs, and car maintenance, can contribute to daily moderate-to-vigorous activity. Activity plans could be individually tailored and designed to align with cultural values. Walking contributed substantially to achieving the daily recommended 30 min of moderate-to-vigorous activity. This result highlights the importance of walkable environments in ethnic neighborhoods. 

## 5. Conclusions

Disaggregating data on Native Hawaiians and Asian subgroups is essential to understand these populations. Our study based on relatively small numbers of participants per year across a range of years gives an indication of physical activity levels. When all types of activity are combined, the study populations meet a minimal recommended standard. More activity, however, came from household activities than from sports and recreation. The time spent undertaking sports and recreation increased when exercising with someone. Programs to increase physical activity should reflect ethnic preferences and consider household activities as well as recreational exercise. 

## Figures and Tables

**Figure 1 healthcare-12-00205-f001:**
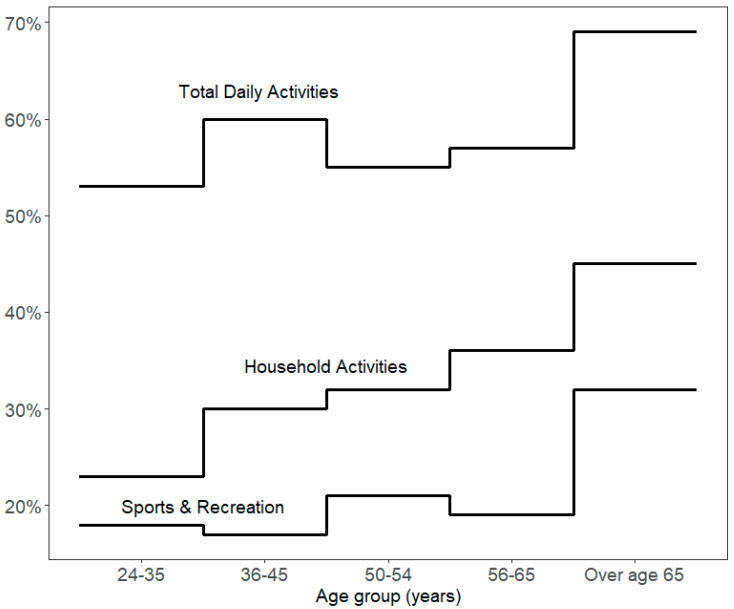
Percent with 30 min or more of moderate-to-vigorous activity by age group and type of activity, ATUS 2010–2019.

**Figure 2 healthcare-12-00205-f002:**
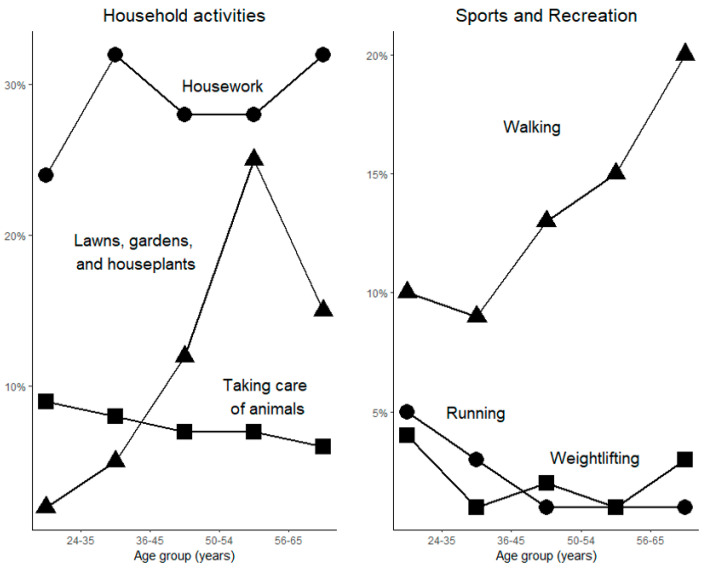
Household activities and sports and recreation by age group. On the left panel circles designate housework, triangles represent activities related to lawns gardens and houseplants, and rectangles indicate taking care of animals. On the right panel triangles designate walking, circles designate running, and rectangles indicate weightlifting.

**Table 1 healthcare-12-00205-t001:** Cross-tabulation of types of moderate-to-vigorous activity by participant characteristics stratified by sports participation and household activities.

Variable	Category	Total	Sports and Recreation		Household Activities	
		N (%)	N (%)	*p* Value	N (%)	*p* Value
Ethnicity	Native Hawaiians	214 (7.1%)	26 (12.2%)	0.038	69 (32.4%)	0.385
	Chinese	767 (25.5%)	189 (24.0%)		233 (26.5%)	
	Asian Indian	600 (20.3%)	130 (20.8%)		200 (28.2%)	
	Filipino	354 (13.6%)	47 (14.7%)		132 (32.1%)	
	Japanese	176 (5.8%)	34 (21.5%)		63 (35.5%)	
	Korean	172 (5.8%)	34 (18.7%)		51 (26.5%)	
	Vietnamese	186 (7.1%)	34 (16.0%)		60 (32.9%)	
	Other Asian	396 (14.8%)	72 (18.7%)		137 (33.2%)	
Age group	Age 24–35	787 (34.3%)	146 (17.7%)	0.001	197 (22.7%)	0.000
	Age 36–45	947 (26.3%)	160 (16.7%)		315 (29.6%)	
	Age 46–55	550 (18.7%)	121 (20.5%)		197 (32.4%)	
	Age 56–65	301 (11.4%)	55 (19.0%)		114 (36.5%)	
	Over age 65	280 (9.4%)	84 (31.9%)		122 (45.0%)	
Sex	Male	1277 (45.9%)	282 (17.3%)	0.015	614 (35.6%)	0.000
	Female	1588 (54.1%)	284 (22.0%)		331 (23.4%)	
Health status	Very good/excellent	854 (54.5%)	184 (23.8%)	0.021	282 (27.8%)	0.400
	Good/fair/poor	672 (45.5%)	127 (17.3%)		233 (30.2%)	
Education	College	1980 (66.4%)	418 (21.5%)	0.025	620 (27.6%)	0.020
	High school	764 (29.4%)	131 (16.0%)		279 (34.8%)	
	Less than high school	103 (4.3%)	15 (15.1%)		36 (33.2%)	
Weekday or weekend	Weekday	1875 (71.4%)	378 (19.5%)	0.892	576 (28.3%)	0.011
	Weekend day	990 (28.6%)	188 (19.3%)		369 (34.3%)	

N’s are unweighted and percentages use survey weights.

**Table 2 healthcare-12-00205-t002:** Odds ratios with 95% confidence intervals of performing 30 min or more of moderate-to-vigorous activities by participant characteristics, stratified by sports participation and household activities.

Variable	Category	Sports and Recreation		Household Activities	
		OR (95% CI)	*p* Value	OR (95% CI)	*p* Value
Race/ethnicity	Native Hawaiians	1		1	
	Chinese	2.24 (1.05, 4.79)	0.038	0.93 (0.64, 1.33)	0.680
	Asian Indian	2.46 (1.13, 5.34)	0.023	1.05 (0.72, 1.52)	0.810
	Filipino	0.83 (0.33, 2.14)	0.707	0.81 (0.54, 1.21)	0.302
	Japanese	1.19 (0.45, 3.12)	0.727	1.10 (0.71, 1.00)	0.668
	Korean	1.95 (0.79, 4.81)	0.146	0.97 (0.58, 1.63)	0.907
	Vietnamese	1.40 (0.57, 3.42)	0.465	0.84 (0.52, 1.34)	0.465
	Other Asian	1.54 (0.69, 3.44)	0.290	1.16 (0.79, 1.70)	0.451
Age group	24–35	1		1	
	36–45	0.80 (0.51, 1.26)	0.339	1.42 (1.08, 1.88)	0.014
	46–55	0.95 (0.57, 1.60)	0.850	1.44 (1.04, 2.00)	0.027
	56–65	1.21 (0.66, 2.22)	0.546	1.78 (1.27, 2.51)	0.001
	Over age 65	3.06 (1.71, 5.49)	0.000	1.96 (1.38, 2.77)	0.000
Sex	Female	1		1	
	Male	1.38 (0.98, 1.95)	0.066	0.66 (0.54, 0.81)	0.000
Health status	Good/fair/poor	1		1	
	Very good or excellent	1.60 (1.12, 2.30)	0.010	1.00 (0.82, 1.21)	0.967
Education	College degree	1		1	
	High school degree	1.23 (0.80, 1.87)	0.343	1.17 (0.93, 1.47)	0.179
	Less than high school	1.18 (0.52, 2.65)	0.694	1.13 (0.75, 1.69)	0.573
Weekday or weekend	Weekday	1		1	
	Weekend	1.13 (0.77, 1.65)	0.540	1.28 (1.05, 1.56)	0.013

OR = odds ratio. CI = confidence interval. 1 indicates the reference.

**Table 3 healthcare-12-00205-t003:** Number (%) of participants and odds ratios with 95% confidence intervals for having 30 min or more of moderate-to-vigorous activity from all activities by participant characteristics.

Variable	Category	N (%)	*p* Value	OR (95% CI)	*p* Value
Race/ethnicity	Native Hawaiians	121 (54.4%)	0.011	1	
	Chinese	488 (58.4%)		1.04 (0.62, 1.74)	0.895
	Asian Indian	386 (57.7%)		1.35 (0.80, 2.28)	0.259
	Filipino	198 (53.4%)		0.58 (0.32, 1.03)	0.063
	Japanese	101 (55.1%)		0.89 (0.45, 1.74)	0.730
	Korean	92 (45.1%)		0.62 (0.31, 1.25)	0.183
	Vietnamese	104 (51.8%)		0.55 (0.29, 1.06)	0.076
	Other Asian	257 (67.2%)		1.33 (0.76, 2.33)	0.323
Age group	24–35	464 (53.3%)	0.005	1	
	36–45	586 (60.0%)		1.26 (0.89, 1.77)	0.190
	46–55	328 (54.6%)		0.99 (0.67, 1.48)	0.974
	56–65	177 (56.7%)		1.44 (0.89, 2.34)	0.139
	Over age 65	192 (69.0%)		2.98 (1.76, 5.07)	0.000
Sex	Female	1004 (59.1%)	0.101	0.83 (0.63, 1.08)	0.168
	Male	743 (54.9%)		1	
Health status	Good, fair, or poor	518 (57.3%)	0.435	1.21 (0.91, 1.60)	0.190
	Very good or excellent	401 (54.6%)		1	
Education	College degree	469 (59.1%)	0.633	1	
	High school	1207 (56.3%)		1.24 (0.90, 1.73)	0.193
	Less than high school	58 (58.3%)		1.29 (0.67, 2.49)	0.442
Weekday or weekend	Weekday	1118 (55.5%)	0.034	1	
	Weekend	629 (61.2%)		1.28 (0.96, 1.69)	0.089

OR = odds ratio. CI = confidence interval. 1 indicates the reference level. N’s are unweighted and percentages use survey weights.

**Table 4 healthcare-12-00205-t004:** Relative minutes with 95% confidence intervals of moderate-to-vigorous activity among participants exercising alone or with someone.

Variable	Category	Mean (SD) in Minutes	*p* Value	RR (95% CI) in Minutes	*p* Value
Exercising with someone	No	111.2 (145.9)	0.558	1	
	Yes	78.1 (73.7)		1.93 (1.63, 2.29)	0.000
Ethnicity	Native Hawaiians	81.5 (83.7)		1	
	Chinese	99.2 (107.5)		0.52 (0.36, 0.76)	0.001
	Asian Indian	97.0 (96.1)		0.58 (0.39, 0.85)	0.006
	Filipino	107.0 (144.5)		0.59 (0.37, 0.95)	0.029
	Japanese	89.8 (150.8)		0.94 (0.61, 1.44)	0.764
	Korean	108.0 (134.7)		0.48 (0.30, 0.78)	0.003
	Vietnamese	94.1 (112.2)		0.58 (0.32, 1.05)	0.072
	Other Asian	92.9 (122.4)		0.72 (0.45, 1.16)	0.177
Age group	24–35	92.2 (117.7)	0.741	1	
	36–45	85.8 (77.4)		1.01 (0.77, 1.32)	0.951
	46–55	79.7 (52.0)		1.02 (0.78, 1.34)	0.874
	56–65	95.5 (114.0)		0.93 (0.68, 1.29)	0.676
	Over age 65	84.9 (95.5)		0.87 (0.67, 1.14)	0.306
Sex	Female	90.0 (110.1)	0.233	1	
	Male	96.2 (102.6)		1.24 (0.99, 1.55)	0.067
Health status	Good/fair/poor	115.5 (156.3)		1	
	Very good or excellent	81.6 (82.2)	0.406	1.12 (0.90, 1.38)	0.300
Education	College degree	123.0 (113.7)	0.017	1	
	High school	60.0 (30.0, 95.0)		1.17 (0.94, 1.47)	0.154
	Less than High school	60.0 (37.3, 122.7)		1.32 (0.87, 1.99)	0.189
Weekday	Yes	60.0 (30.0, 5.0)	0.229	1	
	No	60.0 (37.3, 122.7)		1.42 (1.14, 1.77)	0.002

SD = standard deviation. RR = relative ratio. CI = confidence interval. All the participant characteristics were included in the model estimating the relative minutes of moderate-to-vigorous activity.

## Data Availability

With registration, the ATUS data are publicly available through the Integrated Public Use Microdata Series (www.atusdata.org, accessed on 6 July 2023) (Flood, Sayer, and Backman, 2022 [26]).

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
