# Peer review of "Daily Moderate-to-Vigorous Activity of Native Hawaiians and Pacific Islanders and Seven Asian Subgroups by Types of Activities, American Time Use Survey, 2010–2019"

_healthcare, 2024, doi:10.3390/healthcare12020205_

Round 1

Reviewer 1 Report

Comments and Suggestions for Authors

This study examined MVPA in NHPI and seven Asian ethnic subgroups. This is an understudied group and the authors should be commended for their work. However, the results and discussion need further work – in its current state it is not concise, and much of the discussion simply lists other papers without placing your findings in context. I look forward to the revised version. 

Introduction

There are different fonts in text – please standardise.

Methods

The study design needs more description – was this both rural and urban centres?

Whats the definition of poor quality data

Given Asian ethnicity was not available until 2013, how any NHPI were included prior to 2013?

How did we define MVPA in sports/rec or household activities – please detail

Was financial/employment status recorded

Results

Please given the total n that met inclusion criteria, and the total ATUS in that period of time.

The study variables of health status, education need to be described in methods

What was the overall % of Asian/NHPI in the ATUS?

Ideally, table 2 results should be compared to the rest of the ATUS population. 

Another sensitivity analysis that should be performed is % of MVPA as a percentage of overall time. Previous studies have found that this can influence results significantly

5 tables is too many – many have very similar points to be made as well. Please condense.

Discussion

The first paragraph should clearly delineate the main takeaway points from the study. Please rewrite.

The discussion tends to meander. Keep it concise and relevant to your findings. For example, the final para “The population heath… reduce health inequities” lists a number of studies without placing your findings in context. Much of the discussion is 1-2 line summaries of other studies without placing your findings in context. 

The increase in activity post 60 clearly points towards employment/retirement as a confounder. This should be discussed. See van dyck et al, BMC public health 2016      Longitudinal changes in physical activity and sedentary time in adults around retirement age: what is the moderating role of retirement status, gender and educational level? And Koenemen et al, Do major life events influence physical activity among older adults: the Longitudinal Aging Study Amsterdam IJBNPA 2012

One of the key limitation is use of a subjective measure of MVPA, which has been shown to differ from objective measures such as accelerometry and pedometers – this needs further discussion (please see Foong et al, The association between physical activity and reduced body fat lessens with age—results from a cross-sectional study in community-dwelling older adults, Exp Gerontology 2014). 

Also, time spent in sedentary/light PA was not adjusted for. This is important, particularly given the dose response relationship between MVPA and outcomes such as obesity, muscle mass/strength, BP, etc (see Dose–Response Association Between Physical Activity and Incident Hypertension, Liu et al, Hypertension 2017, Foong et al, Accelerometer‐determined physical activity, muscle mass, and leg strength in community‐dwelling older adults, Journal of cachexia, sarcopenia and muscle 2016, amongst many others 

No need for 2 limitations sections. 

Conclusion

This is poorly written – please revamp. 

Comments on the Quality of English Language

minor gramatical errors

Reviewer 2 Report

Comments and Suggestions for Authors

I have read with interest this paper where the Authors report data regarding the daily moderate-to-vigorous activity of Native Hawaiians and Pacific Islanders and seven Asian subgroups by types of activities. The article is suitable for publication, nevertheless some minor revision is needed.

1.     On page 2, par 2.1. could you please describe the interview technique more precisely?

2.     On page 2, par 2.1. could you please explain how interviewers were recruited and trained?

3.     On page 3, Table 1 are the ethnicities reported in any order? E.g. in numerosity order, percentage order...

4.     On page 10, line 41 please report the percentage in numbers (also for the quotes)

5.      As you report in the “Limitations” paragraph, ATUS is recorded for a single day. Are planning on developing a weekly based survey for future studies? 

Author Response

Reviewer # 2

Comments and Suggestions for Authors

I have read with interest this paper where the Authors report data regarding the daily moderate-to-vigorous activity of Native Hawaiians and Pacific Islanders and seven Asian subgroups by types of activities. The article is suitable for publication, nevertheless some minor revision is needed.

We thank the reviewer for taking the time to read and critique our manuscript. We have made specific changes in the text of the article to address the reviewer’s comments as explained below.

  1. On page 2, par 2.1. could you please describe the interview technique more precisely?

We now provide more detail on the interview technique (Methods, paragraph XXX)

  1. On page 2, par 2.1. could you please explain how interviewers were recruited and trained?

We don’t have information on how the interviewers were recruited by the Bureau of Labor but we do have information on how interviewers were trained. We provide this information in the first paragraph under Study design in the Methods.

  1. On page 3, Table 1 are the ethnicities reported in any order? E.g. in numerosity order, percentage order...

We report NHPI first followed by the Asian ethnicities with the greatest to the least frequency in the data.  We now mention the order in the first paragraph of the Results section.

  1. On page 10, line 41 please report the percentage in numbers (also for the quotes)

            We now report the percentages as suggested.

  1. As you report in the “Limitations” paragraph, ATUS is recorded for a single day. Are planning on developing a weekly based survey for future studies? 

That’s a great idea but since the ATUS is a national survey administered by the Bureau of Labor Statistics we don’t have influence in the survey development.

Round 2

Reviewer 1 Report

Comments and Suggestions for Authors

Many thanks to the authors for comprehensively addressing my comments. I did find this section in the author's reply rather entertaining -

Need to say something without redoing the analyses???

We agree with the reviewer on this point. The focus of the study was time spent in moderate to vigorous activity as recommended by the American College of Sports Medicine to reduce cardiovascular risk. While comparing light exercise to no exercise might also be interesting, it is beyond the scope of this study. We have added a sentence about this point in our discussion (page xx)."

I suppose we all think this but you may wish to not add this in future author's replies! 

But on a more serious note, the reason why this is important is because total amount of time reported affects the reporting of MVPA. Say for example someone reports a total of 10 hours awake, compared to 16 hours awake. This significantly impacts the chance of MVPA >30 min in the latter subject. If total time (which is MVPA + light + sedentary time) is not adjusted for, either by adding it as a variable or using a percentage at least as a sensitivity analysis, the internal validity of the analysis may be affected. This has been found in prior studies. This will need to be added as a major limitation, or a sensitivity analysis (yes, I know we all hate doing more analysis) should be considered if possible. If the sensitivity analysis finds significantly divergent results, it may affect the interpretation of your results. 

Comments on the Quality of English Language

NA

Author Response

We thank the reviewer for the additional comments.  Our intent was to have the manuscript focus on whether the minority ethnicities achieved 30 minutes or more of moderate-to-vigorous activity during the day, a level consistent with national guidelines.  We agree with the reviewer that someone who sleeps for 7 hours a day would have more time to exercise than someone who slept 10 hours a day. We now acknowledge in the limitations that we did not control for sleep and provide the reviewer’s example.
